



# Canopy temperature and heat stress are increased by compound high air temperature and water stress, and reduced by irrigation – A modeling analysis

Xiangyu Luan, Giulia Vico

Department of Crop Production Ecology, Swedish University of Agricultural Sciences (SLU), Uppsala, Sweden

10    *Correspondence to:* Giulia Vico (giulia.vico@slu.se)





**Abstract.** Crop yield is reduced by heat and water stress, and even more when they co-occur. Yet, compound effects of air temperature and water availability on crop heat stress are poorly quantified: crop models, by relying at least partially on empirical functions, cannot account for the feedbacks of plant traits and response to heat and water stress on canopy temperature. We developed a fully mechanistic model coupling crop energy and water balances, to determine canopy temperature as a function of plant traits, stochastic environmental conditions and their variability; and irrigation applications. While general, the model was parameterized for wheat. Canopy temperature largely followed air temperature under well-watered conditions; but when soil water potential was more negative than -0.14 MPa, further reductions in soil water availability led to a rapid rise in canopy temperature — up to 10 °C warmer than air at soil water potential of -0.62 MPa. More intermittent precipitation led to higher canopy temperatures and longer periods of potentially damaging crop canopy temperatures. Irrigation applications aimed at keeping crops under well-watered conditions could reduce canopy temperature, but in most cases were unable to maintain it below the threshold temperature for potential heat damage; the benefits of irrigation became smaller as average air temperature increased. Hence, irrigation is only a partial solution to adapt to warmer and drier climates.

**Keywords**

Canopy temperature; heat stress; water stress; compound events; wheat; irrigation; soil water balance; canopy energy balance





## 1. Introduction

High and stable crop yield requires suitable climatic conditions throughout the growing season. Abiotic stressors, like water scarcity and high temperatures, can adversely affect crop growth, development, and yield, as shown by controlled-condition and field experiments, large scale surveys, and crop model applications (e.g., Zampieri et al., 2017; Daryanto et al., 2017; Kimball et al., 2016; Ray et al., 2015; Asseng et al., 2015). Both water and heat stress impair photosynthesis (Way and Yamori, 2014; Lawlor and Tezara, 2009), undermine crop growth (Hsiao, 1973; Hatfield and Prueger, 2015) and reproduction (Prasad et al., 2011), and hasten crop development and leaf senescence (Lobell et al., 2012), although the physiological mechanisms can differ (Fahad et al., 2017). Heat and water stress do not only act independently but have also compound effects on plant phenology and physiology, so that heat stress is more detrimental if co-occurring with water stress (Mahrookashani et al., 2017; Prasad et al., 2011; Suzuki et al., 2014; Cohen et al., in press). Yet, these compound effects of heat and water stress are seldom considered, experimentally and via models (Rötter et al., 2018).

Climate change is projected to increase temperatures and, in many regions, decrease growing season precipitation or lengthen dry spells (IPCC, 2013). Hot and dry summers are becoming more common (Zscheischler and Seneviratne, 2017; Alizadeh et al., 2020) and changes in climate are already reducing and will likely further reduce crop yield and its stability, and ultimately global food security (Challinor et al., 2014; Masson-Delmotte et al., 2018; Moore and Lobell, 2015; Rosenzweig et al., 2014). The frequency and severity of crop heat and water stress is directly affected by air high temperature and low soil water availability, and is indirectly driven by enhanced evapotranspiration from warm temperatures. Nevertheless, how air temperature and precipitation, and their variability interact in defining the occurrence, extent, and duration of crop heat and water stress has not been investigated in detail.

Canopy temperature allows more accurate estimates of the consequences of heat stress on the crop and its yield than air temperature (Gabaldón-Leal et al., 2016; Siebert et al., 2014; Rezaei et al., 2015). Indeed, canopy temperature can deviate from air temperature under field conditions, because of the interplay among plant traits, plant water availability, air temperature and humidity, solar radiation, wind velocity, and the ensuing canopy microclimate (Michaletz et al., 2016; Schymanski et al., 2013). Considering canopy instead of air temperature is particularly important when characterizing the effects of compound heat and water stress, and the mitigating potential of irrigation against heat stress, because canopy temperature can be substantially higher than air temperature under water stress (e.g., Siebert et al., 2014).

Heat stress and damage are the result of complex and interacting plant physiological processes, depending on the temperature reached by the specific organ and the duration of the stress. Crop response to temperature is nonlinear (Porter and Gawith, 1999; Sanchez et al., 2014). Exceeding crop- and phenological stage-specific thresholds can lead to plant tissue damage and halted physiological processes, although the plant can still survive. Also the duration of exposure to high temperatures affects the outcome. For example, the accumulation of high temperature days negatively affected yield in rainfed systems (Schlenker and Roberts, 2009). In the face of increasing variability in the climatic conditions, we need to determine how stochastic precipitation and air temperature combine in determining canopy temperature. Average canopy temperatures and duration of periods above threshold for damage can provide indications on the exposure of crops to potential heat stress.

Irrigation can buffer some aspects of climatic variability and extremes imposed on crop production (Tack et al., 2017; Zhang et al., 2015; Li and Troy, 2018; Vogel et al., 2019). Irrigation directly alleviates water stress by supplementing precipitation. Further, by sustaining the plant's evaporative cooling, irrigation can reduce canopy temperature and hence the consequences of high air temperature (Vogel et al., 2019; Siebert et al., 2017). In other words, by removing water stress, irrigation can also



diminish the occurrence of heat stress. Nevertheless, we lack a quantification of how much irrigation can reduce the effects of unfavorable air temperature and precipitation, and the occurrence of crop heat stress and compound heat and water stress.

Canopy temperature is difficult to measure directly, although it can be estimated indirectly based on thermal imagery (e.g., Still et al., 2019). Models are a powerful tool to explore how canopy temperature changes with growing conditions and plant traits, beyond what is feasible via direct observations in specific experiments. Existing crop canopy temperature models either link canopy to growing conditions via simple empirical relations (e.g., Shao et al., 2019; Neukam et al., 2016) or model explicitly the leaf or canopy energy balance (Webber et al., 2016; Fang et al., 2014; Webber et al., 2017). But, so far, the role

of plant water availability has been included only via semi-empirical corrections even in mechanistic models. For example, actual canopy temperature was calculated based on canopy temperatures under maximum and zero stomatal conductances and a crop water stress index (see Webber et al., 2018; Webber et al., 2017, for a review of approaches and their performance). Mechanistic models fully representing plant physiology can estimate crop canopy temperature that better reflects soil water and weather dynamics, and how plants respond to environmental conditions. Such models are currently lacking, but are

necessary to quantify the effects of changes in air temperature and precipitation patterns; and the benefits of irrigation.

We developed a mechanistic model to estimate crop canopy temperature as a function of crop physiology, soil features, and (stochastic) climatic conditions, coupling the canopy energy balance and the water transport through the soil-plant-atmosphere continuum (SPAC), with stomatal conductance based on an optimality principle. We used the model in a case study – wheat grown in a temperate climate – to answer the following questions:

-   What are the compound effects of soil water availability and air temperature on crop canopy temperature?
-   How does precipitation pattern influence canopy temperature and its variability, and the duration of potentially damaging canopy temperatures?
-   How effective is irrigation in reducing canopy temperature, depending on the climatic regime?

## 2.  Methods

### 2.1  Model description

To quantify the compound effects of air temperature and precipitation regimes on canopy temperature, and the potential of irrigation to reduce the occurrence of heat stress, we developed a process-based model describing the coupled canopy energy and water balances, and their interactions with the water balance of the rooting zone. See the model structure in Fig 1 and the Supplementary Information – SI – for details and symbols. The model allows exploring how plant traits and physiological

responses to growing conditions interact with air temperature and soil water availability in defining canopy temperature, while relying on parameters with clear physiological meanings (Table S2).

To limit parameter and computational requirements, a minimalist approach was used, lumping the canopy in a 'big leaf' (Amthor, 1994; Jarvis and McNaughton, 1986; Bonan, 2019) and the soil water dynamics in a 'bucket-filling' model, with instantaneous losses via runoff and percolation below the rooting zone (e.g., Milly, 1994; Rodriguez-Iturbe et al., 1999). These

simplifications are expected to have minor repercussions on our conclusions (see SI, Section S5).

As detailed in the SI, combining the canopy water and energy balance, the canopy temperature, $T_c$, can be obtained as





$$T_c = T_a + \frac{Q^\downarrow + B^\downarrow_{n,ref} - \lambda g_{v,c} D}{c_p g_{H,c} + \lambda g_{v,c} s_s + 4\, \varepsilon_c \sigma T_a^3 \left[1 - \exp\left(-K_{bl,d}\ L_{Al}\right)\right]}. \tag{1}$$

where $T_a$ is the air temperature; $Q^\downarrow$ the net absorbed short-wave radiation; $B^\downarrow_{n,ref}$ the net absorbed long-wave radiation at $T_a$ (isothermal radiation); $D$ the atmospheric vapor pressure deficit; $g_{v,c}$ and $g_{H,c}$ the total canopy conductances to water vapor and heat respectively, which include stomatal and aerodynamic conductances; $\lambda$, $c_p$, $\varepsilon_c$, $\sigma$, and $K_{bl,d}$ are constants (Table S1);
$s_s$ the slope of the vapor pressure vs. temperature curve, dependent on $T_a$; and $L_{Al}$ the leaf area index.

We explicitly included the dependence of stomatal conductance on environmental conditions and plant physiology exploiting an optimality principle: plants are assumed to maximize carbon uptake over a given period, subject to limited water availability (Mäkelä et al., 1996; SI, Eq. S9-S11). We chose this approach because it is simple yet based on an evolutionary principle; and has led to promising results (Buckley et al., 2017; Eller et al., 2020). Differently from other stomatal optimization models, the
original Farquhar et al. (1980) model for the photosynthetic rate was approximated with a hyperbolic function that links RuBisCO and electron transport rate limitations while retaining the same physiological parameters of Farquhar's model (Vico et al., 2013). This model was further developed here to account for the effects of the leaf boundary layer conductance and day respiration, and the key stomatal and non-stomatal effects of limited water availability on marginal water use efficiency and metabolic activity (Zhou et al., 2013; Manzoni et al., 2011; Vico and Porporato, 2008; see SI Section S1.2.1 for details). The
results obtained with an alternative, empirical model of canopy conductance parameterized with eddy covariance data (SI, Eq. S30-S32; Novick et al., 2016) further support our mechanistic approach. Those results highlighted the need to explicitly represent canopy gas exchanges to capture the dependence of canopy temperature on air temperature, unless site- and crop-specific data are available to determine the canopy conductance empirically (SI, Fig. S6). Finally, aerodynamic conductances to heat and vapor were determined based on wind velocity, $U$, and leaf width, via semi-empirical relations describing heat and
mass transport inside the leaf boundary layer and to the bulk atmosphere (SI, Sections S1.2.2 and S1.2.3).

The canopy conductances affect and are affected by the soil water balance and water transport along the SPAC. On the one hand, soil water potential influences leaf water potential and hence leaf physiological activities (stomatal conductance, metabolic rates, and marginal water use efficiency). On the other hand, stomatal conductance and atmospheric water demand drive the rate of canopy water losses and hence the decline of soil water content. We represented the soil water content as soil
saturation, $s$ ($0 \leq s \leq 1$; soil moisture hereafter), linked to soil water potential, $\psi_s$, via texture-dependent soil water retention curves (SI, Eq. S24). A bucket-filling model was used to describe the soil moisture dynamics, with precipitation and irrigation as input and evapotranspiration, deep percolation below the rooting zone and superficial runoff as losses, but neglecting the root structure, the time needed for the water to be redistributed within the soil, and lateral soil water movements (SI Section S1.3.1; Vico and Porporato, 2010). The soil water balance was coupled to a minimalist description of water transport through
the SPAC, to determine the leaf water potential. The SPAC was modeled as a series of conductances, from the soil, through the plant, to the atmosphere (SI, Section S1.3.2; Manzoni et al., 2013).

These model components provide conductances and boundary conditions to apply Eq. (1) and quantify how canopy temperature, $T_c$, changes with environmental conditions and management: air temperature and humidity, wind velocity, incoming solar radiation, and precipitation; and irrigation applications, if any. The model needs to be solved iteratively (Fig.
1). At each time step (a day; see Section 2.2), the model considers the previous soil moisture and current atmospheric conditions. The previous canopy temperature and water potential are used as initial guesses for the numerical integration. First, the model determines the canopy boundary layer and aerodynamic bulk conductances and water supply and demand. Then, canopy water potential $\psi_c$ is determined iteratively by equating water supply and demand. After convergence is reached on





$\psi_c$, the canopy energy balance is used to determine iteratively $T_c$. Finally, the soil water balance is updated with inputs and

losses cumulated over the time step.

Based on $T_c$, we derived two metrics representing the potential for heat stress damage: i) $T_{c,mean}$, the mean canopy temperature during a specific period (anthesis; see Section 2.2); and ii) $P_{CHS}$, the fraction of days during such period when $T_c$ exceeded the crop-specific threshold $T_{th}$, above which detrimental effects of crop heat stress are likely. $P_{CHS}$ is thus a measure of the duration of the detrimental conditions, while $T_{c,mean}$ quantifies the level of detrimental conditions.

**2.2  Case study**

While the model is of general applicability, we focused on the case of wheat (*Triticum aestivum*) – a staple crop with relatively low tolerance to high temperatures when compared with other crops (Sanchez et al., 2014) – grown at 45 ° latitude N. All the model parameters are summarized in the SI, Table S2.

We restricted our analyses to anthesis, when wheat is most vulnerable to heat (Porter and Gawith, 1999) and water (Daryanto

et al., 2017) stress. Anthesis was assumed to last 21 days (Mäkinen et al., 2018), starting at 140th day of the year (in line with observations and simulations at the latitude selected; Semenov et al., 2014; Bogard et al., 2011). For simplicity, the timing and length of anthesis was kept constant under all air temperature scenarios.

The model is capable of simulating the diurnal course of the key variables, but, for simplicity, we focused on the central part of the day, when incoming short-wave radiation at the top of the canopy $Q_0^{\downarrow}$ and air temperature $T_a$ are at or near their daily

maxima, and $T_c$ is expected to peak. Wind velocity $U$ was assumed to be at the lowest end of its realistic range and $Q_0^{\downarrow}$ that of clear sky conditions, thus providing the maximum expected $T_c$ and a conservative estimate of the frequency of occurrence of potentially damaging temperatures.

While the model can be driven by measured environmental conditions, to systematically explore several climate scenarios, we employed synthetically generated environmental conditions. Daily precipitation was idealized as a marked Poisson process

(Rodriguez-Iturbe et al., 1999), i.e., exponentially distributed interarrival times, with average frequency $\lambda_p$. Event depth was also assumed to be exponentially distributed, with average $\alpha_p$ (SI, Section S1.4.2). The variability of $T_a$ around its long-term average $\mu_{T_a}$ was described via an Ornstein-Uhlenbeck process (see Section S1.4.3 in the SI; Benth and Benth, 2007). In line with the focus on the warmest part of the day, $T_a$ is to be interpreted as the maximum daily air temperature. Finally, $U$, $Q_0^{\downarrow}$, and $RH$ were assumed to be constant during the simulations (SI, Table S2), whereas air water vapor pressure, $e_a$, and vapor

pressure deficit, $D$, were calculated based on $T_a$ (Campbell and Norman, 1998).

As baseline pedoclimatic conditions, we considered a sandy loam soil, average precipitation frequency $\lambda_p$ of 0.2 d$^{-1}$, average event depth $\alpha_p$ of 8.2 mm (corresponding to an average annual precipitation total of 600 mm), long-term average air temperature $\mu_{T_a}$ of 25 °C, air temperature standard deviation of 3.6 °C, air relative humidity $RH$ of 40%, wind velocity $U$ of 4 m s$^{-1}$, and net incoming short-wave radiation $Q_0^{\downarrow}$ of 800 W m$^{-2}$. We also explored additional pedoclimatic conditions.

Specifically, we considered more extreme precipitation scenarios, comprising increasing precipitation due to increasing precipitation frequency; and a constant average annual precipitation total, but more intermittent precipitation, with reduced average precipitation frequency ($\lambda_p$=0.07 d$^{-1}$) and increased average event depth ($\alpha_p$=23.5 mm). Long-term average air temperature $\mu_{T_a}$ also of 20 and 30 °C were explored. Separate sensitivity analyses were run for the standard deviation of air temperature (SI, Fig. S3), soil texture (SI, Fig. S4), and $U$, $Q_0^{\downarrow}$, and $RH$ (SI, Fig. S5).



For the irrigated case, a demand-based (water) stress-avoidance irrigation was considered, whereby an irrigation application is triggered whenever soil water potential reached the intervention point, $\tilde{\psi}_s$ (Vico and Porporato, 2011). To ensure well-watered conditions, $\tilde{\psi}_s$ was set to -0.07 MPa, i.e., just above the incipient water stress for wheat (-0.1 MPa; Kalapos et al., 1996). Each irrigation application restored a pre-set target soil water potential, $\hat{\psi}_s$.

Finally, the crop- and phenological-stage specific temperature threshold above which detrimental effects of crop heat stress

are likely, $T_{th}$, was set equal to the maximum baseline temperature during anthesis. $T_{th}$ is a large source of large uncertainty, when aiming at defining the occurrence of crop heat stress and its consequences on the crop and final yield (Siebert et al., 2017; Wanjura et al., 1992). Even within a specific developmental stage, there is a large variability of reported baseline and optimal temperatures, because of differences in variety, growing conditions, and experimental approach. Further, crop's baseline and optimal temperatures are often defined based on air temperature, although plants respond to canopy or even organ

temperature. As discussed below, the differences between air and canopy temperatures can be large, in particular under limited plant water availability. To make the comparison between $T_c$ and $T_{th}$ meaningful, here we considered a maximum baseline temperature obtained under well-watered conditions and low $D$; and set $T_{th}$ equal to 30 °C (Saini and Aspinall, 1982). This value is in agreement with those obtained in other experiments focusing on wheat (Porter and Gawith, 1999).

### 2.3  Statistical tests

The simulated canopy temperatures were not normally distributed, according to the Anderson-Daring test ($p<0.05$). Hence, to test if median $T_{c,mean}$ and $P_{CHS}$ differed across scenarios, we employed the Mood's test; and to test difference in their variances, we used the Brown-Forsythe's test. The test results are summarized in SI Tables S3-S8. Differences are commented on when p<0.05.

### 3.   Results

The stochasticity of air temperature, $T_a$, and precipitation occurrence was mirrored by the erratic variations of soil moisture, $s$, and canopy temperature, $T_c$, in the numerically simulated trajectories (exemplified in Fig. 2). $T_c$ largely followed $T_a$, but $s$ determined whether $T_c$ was near or above $T_a$. Under well-watered conditions, when $s$ ensured unconstrained transpiration, $T_c$ was similar to or even occasionally lower than $T_a$; whereas when $s$ decreased, $T_c$ became warmer than $T_a$ (after approximately day 12 in Fig. 2). The evolution of $T_c$ and other key physiological state variables, including stomatal conductance, photosynthesis and canopy water potential, during a dry down is reported in the SI, Fig. S1.


Despite the complex mechanisms linking $T_a$ and plant water availability to $T_c$, the resulting temperature difference $T_c - T_a$ followed a relatively simple pattern (Fig. 3). When $s$ was above 0.34 (corresponding to $\psi_s$=-0.14 MPa for the soil chosen), $T_c$ was within 1 to 2 °C of $T_a$, with $T_c < T_a$ for $T_a$>25 °C. Conversely, for $s$<0.34, $T_c - T_a$ increased as $s$ declined, with increasing slope, from 1 °C at $s$=0.34 to 10 °C at $s$=0.25 (corresponding to $\psi_s$=-0.62 MPa); $T_c - T_a$ was independent of $T_a$

(i.e., under water stress $T_c - T_a$ is driven by soil water availability for evaporative cooling). Hence, high $T_c$ could be caused by high $T_a$  or low $s$ or their combination. The dependence of the plant's physiological state variable on $s$ is reported in the SI, Fig. S2, for set $T_a$.

Temperature and precipitation patterns interacted in defining the mean temperature during anthesis, $T_{c,mean}$. Increasing average precipitation totals decreased median $T_{c,mean}$ (colors in Fig. 4, SI Table S3, S4), in particular at lower precipitation

totals (red in Fig. 4) and higher long-term mean air temperature $\mu_{T_a}$ (right in Fig. 4). $T_{c,mean}$ was less affected by annual



average precipitation totals larger than 900 mm and $\mu_{T_a}$ at 20 °C. $T_{c,mean}$ variability increased with $\mu_{T_a}$ and, to a lesser extent, with decreasing average precipitation totals (SI Table S3, S4).

Precipitation regime affected median of $T_{c,mean}$ and its variability even when considering the same precipitation total but different average precipitation frequencies, $\lambda_p$ (and hence event depths, $\alpha_p$; Fig. 5, top). When compared with the baseline precipitation scenario (red bars), larger but more intermitted events (i.e., lower $\lambda_p$ and higher $\alpha_p$; violet bars) resulted in higher $T_{c,mean}$ median and variability in rainfed cropping (SI, Table S5). The median of $T_{c,mean}$ increased with $\mu_{T_a}$ regardless of rainfall pattern, whereas the variance increased in the baseline rainfed scenario, except under more intermittent precipitation (Table S6).

Irrigation reduced median and variance of $T_c$ with respect to rainfed cropping under the same climatic scenario (red vs. blue hues in Fig. 5, top). Also the dependence of $T_c$ on precipitation pattern was reduced with irrigation (SI, Table S5). Yet, even with irrigation, median and variability of $T_c$ increased with $\mu_{T_a}$, although such increase was less marked than that under rainfed cropping (SI, Table S6).

Irrigation applications reduced the fraction of days during which $T_c$ was above the threshold temperature for potential heat damage, $T_{th}$, i.e., of likely crop heat stress ($P_{CHS}$; Fig. 5 bottom). But it could not completely prevent this occurrence (i.e., median $P_{CHS} > 0$), except for $\mu_{T_a}$=20 °C. Among the climatic scenarios considered, the largest mean reduction in $P_{CHS}$ (100%) occurred at $\mu_{T_a}$= 20 °C, and the smallest (50%) in the more intermittent precipitation scenario at $\mu_{T_a}$=30 °C (Table 1).

Increasing air temperature variability left median and variance of $T_{c,mean}$ unaltered in rainfed cropping, but increased them in irrigated cropping (SI, Fig. S3 top and Table S7). There, the removal of water stress via irrigation made the resulting canopy temperature more sensitive to the air temperature regime. The median and variance of $P_{CHS}$ increased with temperature variability, except the variance under the more intermittent rainfed scenario (Fig. S3 bottom, Table S7). Finer soil texture did not affect $T_{c,mean}$ and $P_{CHS}$, although the difference between rainfall scenarios remained (Fig. S4 and Table S8). Also incoming short-wave radiation $Q_0^\downarrow$, wind velocity $U$, and air relative humidity $RH$ affected $T_c$ (Fig. S5). An increase of $Q_0^\downarrow$ increased $T_c$, in particular at $s<$ 0.35. Decreasing $U$ enhanced $T_c$ for $s<$ 0.35, but did not affect it when $s>$0.35. In contrast, $T_c$ slightly increased with $RH$ for $s>$0.35, but showed no response to it when $s<$ 0.35.

## 4. Discussion

### 4.1 Soil water availability and air temperature jointly affect canopy temperature

We quantified the compound effect on canopy temperature of environmental conditions: air temperature, soil water availability, incoming short-wave radiation, wind velocity, relative humidity, soil texture, and irrigation. Our model is an improvement with respect to existing approaches to simulate canopy temperature in agricultural systems, which rely on empirical corrections of values determined by means of the energy balance under extreme conditions (Fang et al., 2014; Webber et al., 2016). Lacking adequate modelling tools has limited our ability to effectively quantify the likelihood and extent of potential heat damage.

The role of environmental conditions is mediated by plant physiology and its response to conditions. But, despite the complex mechanisms behind canopy temperature, the resulting pattern was relatively simple. Canopy temperature increased from cooler



temperatures and wetter soils to warmer and drier conditions (Fig. 3). Under well-watered conditions, some thermoregulation occurred, cooling down or warming up the canopy depending on the air temperature, to ensure near optimal temperature for photosynthesis (Michaletz et al., 2016); this thermoregulation capability was lost when water limited evaporative cooling. The differences of canopy and air temperatures obtained with the model are in line with experimental observations and other model results, thus lending support to our approach. In wheat, for example, daily maximum or mid-day canopy temperature was 2 to 10 °C higher than air temperature under water stress, and up to 6°C cooler than air temperature under well-watered condition, among field observations and model results (Pinter et al., 1990; Rashid et al., 1999; Jensen et al., 1990; Howell et al., 1986; Ehrler et al., 1978; Balota et al., 2008; Neukam et al., 2016; Webber et al., 2016). Our simulations led to canopies being 2 to 10 °C warmer than air under water stress, and to a cooling effect of 1 to 2 °C under warm but well-watered conditions. Differences between model results and observations can be ascribed to cultivar-specific traits, approach to measuring canopy temperature, measurement timing and position (within or just above the canopy), and environmental conditions (e.g., solar radiation, soil texture). Some of these aspects can be accounted for by the model, by adjusting the parameters to the specific crop and variety and environmental conditions.

The difference between canopy and air temperature was higher than, and independent of, air temperature when soil water potential was below a critical value (Fig. 3). This threshold-like response mirrors that of stomatal closure and plant transpiration reduction with water stress (for wheat, e.g., Sadras and Milroy, 1996; Shen et al., 2002; Wang et al., 2008; Wu et al., 2011; Kalapos et al., 1996). Yet, no threshold for stomatal closure was imposed *a priori* in the model. The emerging threshold of soil water potential (-0.14 MPa) is comparable with the soil water potential corresponding to incipient stomatal closure in some experiments (-0.1 MPa; Kalapos et al., 1996), but lower than those of others (between -0.27 and -0.35 MPa depending on the cultivar; Wang et al., 2008) and higher than the value often assumed to correspond to well-watered conditions (-0.03 MPa; Ali et al., 1999; Laio et al., 2001).

### 4.2 More intermittent precipitation and higher air temperature increase canopy temperature

Climate change is expected to alter both air temperature and precipitation regimes, with further increases in average and extremely high air temperatures, and, in some regions, scarcer or more intermittent precipitation, i.e., longer dry spells (IPCC, 2013). Co-occurring dry and hot extremes are becoming increasingly frequent (Alizadeh et al., 2020; Zscheischler and Seneviratne, 2017). We showed that these compound changes can increase canopy temperature and its variability (Fig. 4 and 5).

For set air temperature conditions, even with same average precipitation totals, less frequent but larger precipitation events increased median and variance of canopy temperature, as well as the fraction of days during which the temperature threshold for potential heat damage was exceeded (Fig. 5). Larger, less frequent precipitation events result in enhanced losses via runoff and percolation below the rooting zone, thus reducing plant water availability; the ensuing (longer) dry down can thus lead to lower soil moisture levels, potentially enhancing canopy temperature. This result points to the importance of considering not only seasonal precipitation totals but also their timing. Indeed, reductions in the number of rainy days have already reduced crop yield, and could even override the benefits of increased total precipitation (Ram, 2016).

An increase in average air temperature resulted not only in a higher mean canopy temperature during anthesis, as expected (Eq. 1), but also in a larger variability of such mean (Fig. 4 and 5). Even the extent of changes in mean canopy temperature during anthesis caused by alterations of the precipitation regimes depended on average air temperature (Fig. 5). Intermediate mean air temperature resulted in the largest response of mean canopy temperature to changes in precipitation frequency under constant average precipitation totals (Fig. 5). These complex, compound effects show that it is necessary to explicitly consider not just the means but also the timing and variability of air temperature and precipitation, and their joint effects, when




quantifying the potential of climate change to cause crop heat stress. Hence, models accounting in full for the stochasticity of environmental conditions are needed.

Crops are also faced by increasing air $CO_2$ concentration. While this further global change was not explored here, we speculate that an increase in air $CO_2$ concentration could reduce stomatal conductance and hence enhance canopy temperature, all the other conditions being the same. But reduced stomatal conductance can also reduce the rate of soil water storage depletion and hence the maximum canopy temperature reached during a dry down. The net results of an increase in air $CO_2$ concentration is thus expected to be small. Indeed, air $CO_2$ concentration of 200 to 220 ppm above ambient increased canopy temperature only of 1 °C in Free Air $CO_2$ Enrichment experiments and in model simulations (Webber et al., 2018; Kimball et al., 1999); and a weak reduction of yield loss to heat with enhanced $CO_2$ is expected (Schauberger et al., 2017).

### 4.3 Irrigation reduces but does not cancel the risk of heat stress

By reducing the occurrence and extent of water stress, irrigation could lower canopy temperature, and its variability, as well as the frequency of it exceeding the threshold for potential heat damage (Fig. 5). This reduction was particularly marked at lower average air temperature and under more intermittent precipitation (**Error! Reference source not found.**). Yet, irrigation aiming at maintaining the plants under well-watered conditions could not completely remove the possibility that canopy temperature exceeded the temperature threshold for potential heat damage, except under the coolest air temperature scenario. Further, the benefits of irrigation became smaller as air temperature increased.

The risk of canopy temperature exceeding the temperature threshold for potential heat damage under (water) stress avoidance irrigation can be interpreted as the potential heat stress attributable only to air temperature. This is because no limitation to evaporative cooling is expected under the imposed irrigation scenario, where the soil water potential triggering an irrigation application was less negative that the critical soil water potential emerging from Fig. 3. The reduction of the fraction of time when canopy temperature is above the threshold for potential heat damage obtained via irrigation is a measure of the relative role of air temperature and water stress in defining high canopy temperatures, thus disentangling their relative importance. In addition, for the most effective use of the available resources against heat stress, the emerging threshold of soil water potential that limit water-stress induced high canopy temperatures (Fig. 3) could be used to define a crop-specific irrigation intervention point for irrigation. Maintaining the soil water potential above that threshold would require additional water resources while leading to marginal further cooling effects, i.e., little advantage in staving off heat stress.

Irrigation could not fully eliminate the negative effects of heatwaves and the warmer conditions expected in the future. Even for air temperatures for which irrigation can reduce the potential for heat stress damage, expanding irrigation to mitigate the effects of high canopy temperatures can be unadvisable or impossible, due to physical or economic water scarcity (Rosa et al., 2020), already unsustainable exploitation of water resources (Wada et al., 2010), or the negative impacts of irrigation on soil salt content and nearby water bodies (Daliakopoulos et al., 2016; Scanlon et al., 2007). Other management approaches are thus needed to limit the potential for crop heat stress, in particular under high average air temperatures (Deryng et al., 2011; Lobell et al., 2008). Examples are shifting to more heat-tolerant cultivars and species (Tack et al., 2016); altering the sowing date (Lobell et al., 2014; Mourtzinis et al., 2019) or migrating crops (Sloat et al., 2020) so that anthesis occurs when air temperature is, on average, lower.

### 5 Conclusions

Longer dry spells and high temperatures are expected to become even more frequent in the future, with potential negative and compounded effects on crop development and yield. Exploring the occurrence and severity of crop heat stress requires

quantifying canopy temperature and considering under which conditions it exceeds the temperature threshold known to create appreciable damage. We developed a mechanistic model to determine canopy temperature, based on the explicit coupling of the soil water dynamics with the canopy energy balance and an optimality principle, mechanistically accounting for plant physiology and its response to (stochastic) environmental conditions.

Using wheat as a case study, we explored how canopy temperature and its variability changed with stochastic air temperatures and precipitation, in rainfed and irrigated cropping. When soil water potential was less negative than -0.14 MPa, the additional

benefit of an increase in soil water availability and hence potential evaporative cooling became marginal; and thermoregulation ensured semi-optimal leaf temperature. However, canopy temperature rose rapidly above air temperature when soil water potential was less than -0.14 MPa, due to lowered evaporative cooling.

Less frequent and more intense precipitation caused more variable soil water contents, leading to higher and more variable canopy temperatures, and a higher fraction of days when the temperature threshold for potential heat stress damage was

exceeded. Larger precipitation totals and irrigation applications could reduce the occurrence of high canopy temperature and the potential for heat damage. Yet, irrigation could not completely remove the risk of crop heat stress when long-term mean air temperature was 25 °C or higher, calling for alternative management solutions.

Accurate estimates of canopy temperature are necessary to assess the role of precipitation and air temperature patterns in defining the risk of heat stress, and evaluate the mitigation potential of irrigation. Mechanistic models explicitly linking plant

physiology to environmental conditions also allow exploring the effects of plant traits on the occurrence and extent of water and heat stress. As such, these models can support management decisions, from irrigation applications to identifying crops able to avoid heat stress.

*Code availability*. The code is available upon request, by contacting the corresponding author.


*Data availability*. Data for model parameterization are available in the cited literature.

*Author contributions*. GV conceived the idea. XL and GV developed the codes of the model. XL performed the analyses and created the figure. XL and GV wrote the manuscript. GV revised the manuscript.


*Competing interest*. The authors declare that they have no competing interests.

*Special issue statement*. This article is submitted to the special issue "Understanding compound weather and climate events and related impacts".


*Acknowledgments*. We thank Maoya Bassiouni for feedback on the manuscript. The support of the Swedish Research Council (Vetenskapsrådet), under grant 2016-04910, is gratefully acknowledged. GV also acknowledges the partial support of the





project COSY funded by Swedish Research Council for Sustainable Development (FORMAS, under grant 2018-02872); and
the project iAqueduct, within the 2018 JPI Joint Programming Initiative Water challenges for a changing world - Water Works
2017 ERA-NET Cofund, through FORMAS grant 2018-02787.

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




**Table 1 Reduction in the potential for heat stress by irrigation, as summarized by the mean reductions of $P_{CHS}$ from rainfed cropping to stress avoidance irrigation, using rainfed as reference.**

| $\mu_{T_a}$ (°C) | Baseline precipitation regime ($\alpha_p$ =8.2 mm ; $\lambda_p$ =0.2 d⁻¹) | More intermittent precipitation ($\alpha_p$ =23.5 mm ; $\lambda_p$ = 0.07 d⁻¹ ) |
|---|---|---|
| 20 | 100% | 100% |
| 25 | 78% | 82% |
| 30 | 53% | 50% |


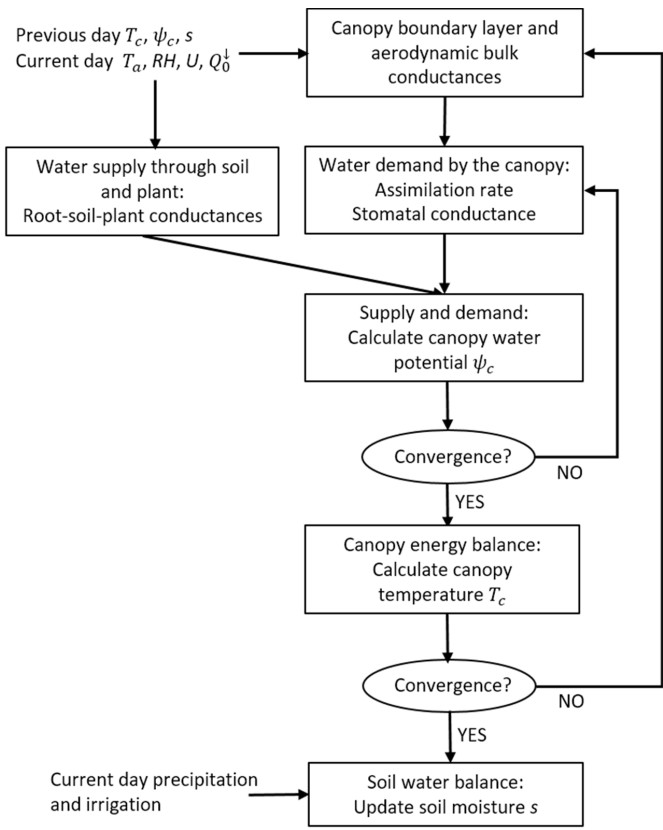

**Figure 1. Flow diagram of the determination of canopy temperature and soil moisture dynamics.**






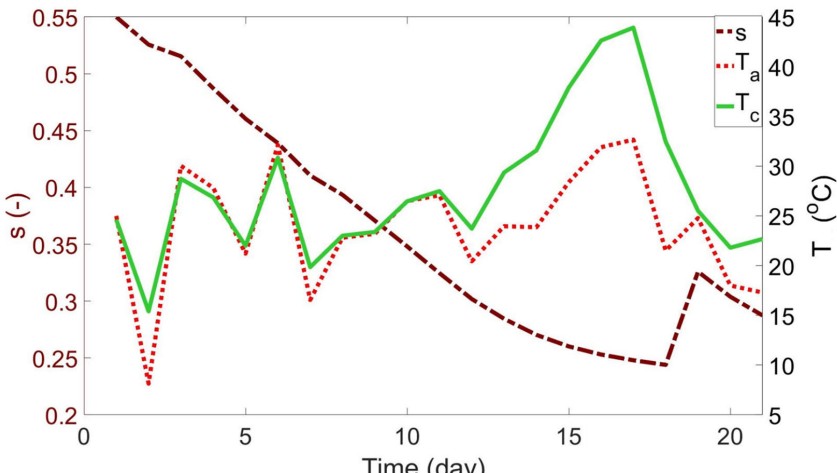

**Figure 2. Example of numerically generated time series of soil moisture ($s$; dot-dashed burgundy line), air temperature ($T_a$; dotted red line), and canopy temperature ($T_c$; solid green line), for rainfed cropping. The left axis represents soil moisture, the right axis temperature. The model was run for 21 days with the baseline environmental conditions. Parameter values are listed in Table S2.**


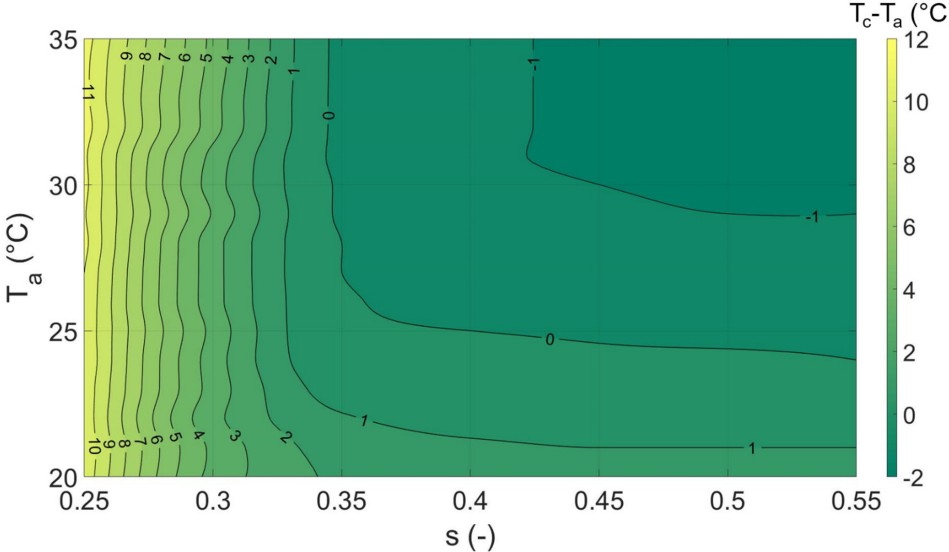

**Figure 3. Canopy-air temperature difference, $T_c - T_a$ (colors and contour lines), as a function of soil moisture ($s$; x-axis) and air temperature ($T_a$; y-axis) for a sandy loam. All other parameters are summarized in Table S2.**




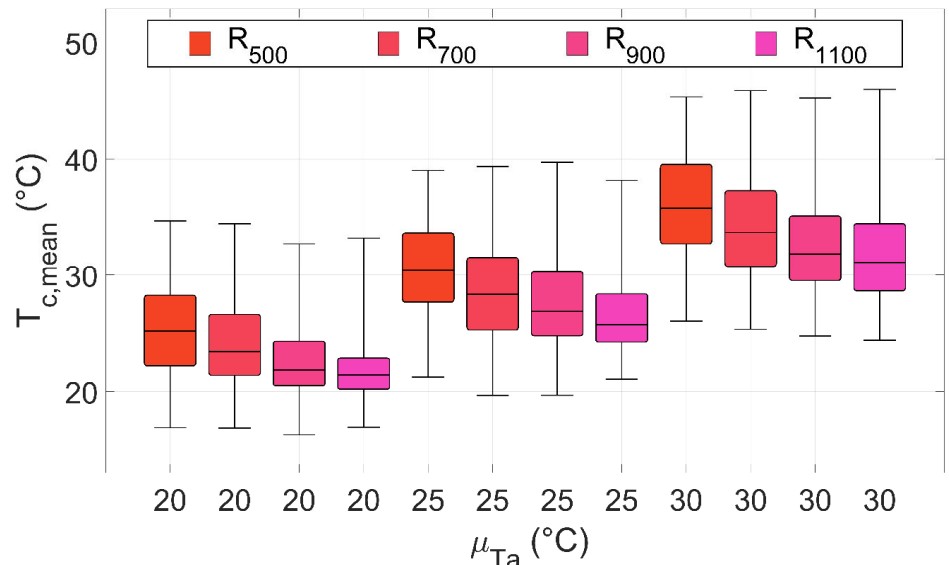

**Figure 4. Mean canopy temperatures during anthesis, $T_{c,mean}$, for four average annual precipitation totals (500, 700, 900, 110 mm; colors) and three long-term average air temperatures $\mu_{T_a}$ (20, 25 and 30 °C; x-axis) Average precipitation depth $\alpha_p$ was kept at 15 mm, while average precipitation frequency $\lambda_p$ changed within each group of 4 boxes, from 0.091 to 0.137, 0.183, and 0.228 $d^{-1}$ (left to right), leading to increasing average annual precipitation totals (subscripts in the legend). For each climatic scenario, 500 21-day simulations were run. The horizontal black lines are the median values; the boxes extend from the first to the third quartile; whiskers cover the whole range.**

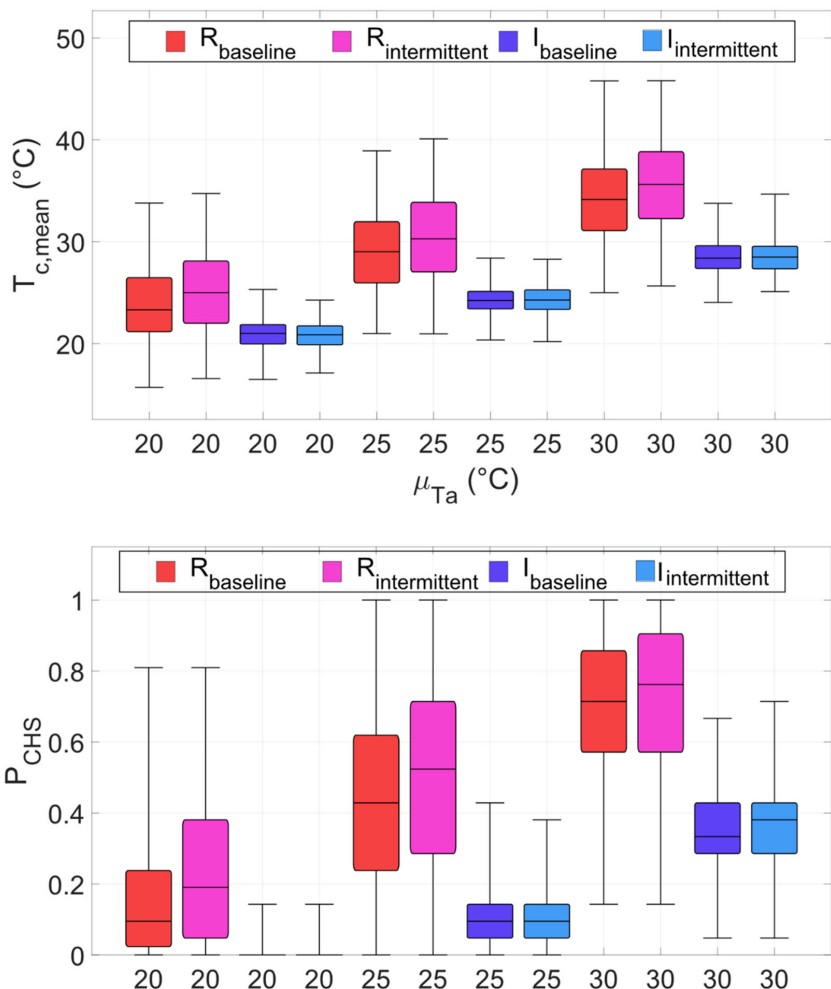


**Figure 5.** Mean canopy temperature during anthesis ($T_{c,mean}$; top) and percentage of days during which $T_c$ is above the threshold temperature for potential heat damage, $T_{th}$ ($P_{CHS}$; bottom), under three long-term average air temperatures $\mu_{T_a}$ (x-axis) and different precipitation and irrigation scenarios (colors). R$_{baseline}$ and R$_{intermittent}$ represent rainfed cropping, respectively under baseline precipitation ($\alpha_p$ =8.2 mm ; $\lambda_p$ =0.2 d$^{-1}$) and more intermittent precipitation ($\alpha_p$ =23.5 mm ; $\lambda_p$ = 0.07 d$^{-1}$). I$_{baseline}$ and
I$_{intermittent}$ refer to stress avoidance irrigation, under the same precipitation regimes of the corresponding rainfed case. For each climatic scenario, 500 21-day simulations were run. The horizontal black lines are the median values; the boxes extend from the first to the third quartile; whiskers cover the whole range.