# Peer review of "S1 Model description"

_Hydrology and Earth System Sciences, 2020_

## Referee Comment (RC1) · Anonymous Referee #1 · 4 Dec 2020

The manuscript presents a new model analysis of the combined effect of water and heat stress. The results are based on the parametrization of wheat during anthesis. The manuscript is well written and structured. The new model formulation and results are relevant to a broad audience. The supplementary describes the model formulation and assumptions in great detail with appropriate references.

The major points to reconsider are the hydrology of the model and the irrigation. The results suggest that the temperature difference ($T_c-T_a$) is sensitive to the soil moisture value. The main nonlinearity seems to be at around 0.33 (Fig. S2).

The drainage and runoff are handled by removing instantly all water above the s1 value. Is this a good assumption during all the different precipitation scenarios and irrigation? It may be that discarding instantly runoff and drainage and not including any ponding may not be realistic in the high precipitation case (P=1100 mm) for 30 cm root-zone depth. What was the ratio of drainage and runoff to precipitation in the different simulations? Have the authors considered modeling the drainage using for example hourly time step when soil moisture exceeds the field capacity? Or some other more explicit approach that might increase the time soil stays wet? Could a more explicit approach to drainage, runoff and ponding result in a larger difference in the results between the soil textures? Allowing drainage and runoff to operate for a longer time may be important when analyzing the 21 day period.

The irrigation is triggered whenever soil water potential reached the intervention point which is just above the water stress point of wheat. It would be nice to know what was the resulting average irrigation frequency with this strategy? Is this frequency similar to the typical wheat irrigation frequency? Was the target soil water potential for irrigation optimized to result in typical wheat irrigation frequency? Report also the amount of irrigation and its ratio to precipitation in the result section.

Specific comments:

L45: "are directly affected high air temperature"

L98: Should you rephrase this question? The results and discussion seem to only report that irrigation cannot completely remove the heat stress but there seems to be no quantification of this effect?

L127: In what sense superficial? Perhaps, surface runoff?

L166: What is the basis of the precipitation parameters? Is intermittent case based on some estimate for some region?

L298: Reference missing.

---

## Author Comment (AC1) · 17 Dec 2020

**https://doi.org/10.5194/hess-2020-549**
**Response to Referee # 1**

**In italics the original comments; in normal font our responses.**

*The manuscript presents a new model analysis of the combined effect of water and heat stress. The results are based on the parametrization of wheat during anthesis. The manuscript is well written and structured. The new model formulation and results are relevant to a broad audience. The supplementary describes the model formulation and assumptions in great detail with appropriate references.*

We thank the Referee for their supportive comments.
As detailed below, should we be offered the possibility to revise our manuscript, the main changes we propose are:
- i) to better justify the approach to the modelling of the soil water balance, and the parameterization of irrigation;
- ii) to report information on the system's water fluxes, under the different precipitation and irrigation scenarios;
- iii) to clarify in which sense the results presented in the manuscript can quantify the role of irrigation in reducing the risk of heat damage.

*The major points to reconsider are the hydrology of the model and the irrigation. The results suggest that the temperature difference (Tc-Ta) is sensitive to the soil moisture value. The main nonlinearity seems to be at around 0.33 (Fig. S2).*
*The drainage and runoff are handled by removing instantly all water above the s1 value. Is this a good assumption during all the different precipitation scenarios and irrigation? It may be that discarding instantly runoff and drainage and not including any ponding may not be realistic in the high precipitation case (P=1100 mm) for 30 cm rootzone depth. What was the ratio of drainage and runoff to precipitation in the different simulations? Have the authors considered modeling the drainage using for example hourly time step when soil moisture exceeds the field capacity? Or some other more explicit*
*approach that might increase the time soil stays wet? Could a more explicit approach to drainage, runoff and ponding result in a larger difference in the results between the soil textures? Allowing drainage and runoff to operate for a longer time may be important when analyzing the 21 day period.*

Stimulated by the Referee's comment, we have investigated in depth the implications of the simplifying assumptions related
to the soil water balance. Specifically, we compared two approaches for the modelling of the losses via surface runoff and deep percolation.
The simplest approach is the one we had employed in the original submission, assuming an instantaneous loss of water whenever inputs would bring soil moisture above the threshold $s_1$. The advantage of this approach is that it can be run at the daily time scale, i.e., at the same scale at which the leaf temperature model is currently run, without incurring in numerical
issues stemming from the nonlinear dependence of soil water losses at high soil water contents.
The second approach considers runoff from saturation excess (but not from infiltration excess), and the nonlinearity of soil hydraulic conductivity and hence percolation below the rooting zone. In this case, the losses via deep percolation are assumed to change with soil moisture as (Clapp and Hornberger, 1978)

$$LQ = K_{sat} s^{2b+3},$$

where $K_{sat}$ is the soil hydraulic conductivity at soil saturation and $b$ is an empirical exponent (the same used to link soil moisture and soil water potential; Eq S24 in the original submission). For sandy loam, $K_{sat}$=0.8 m d$^{-1}$ and $b$=4.90. This approach is more mechanistic than the one based on the threshold $s_1$, since it explicitly considers the soil hydraulic
conductivity and its dependence on soil texture and moisture content. The disadvantage is that, due to the high nonlinearity of the hydraulic conductivity, it requires being solved at a subdaily time scale to avoid large numerical errors (in the analyses below, the integration step was set to 15 min; shorter steps do not alter our conclusions).

To focus on the effects of the soil water balance model and avoid any potential confounding effect (e.g., due to differences in plant water uptake due to different soil moisture levels), we performed this model comparison assuming a fixed dependence of soil water uptake on soil moisture, of the form

$$ET = \begin{cases} ET_{max} & s \geq s^* \\ ET_{max} \dfrac{s}{s^*} & s < s^* \end{cases},$$

where $ET_{max}$ is the transpiration rate under well-watered conditions and $s^*$ is the soil moisture level corresponding to incipient stomatal closure. This dependence is in line with that emerging from the full model (Fig. S2d in the submitted manuscript). In the analyses below, these values were estimated based on the full model output. Specifically, we set $ET_{max} = 4$ mm d$^{-1}$, which corresponds to a surface conductance $G_s$ of 0.39 mol m$^{-2}$ s$^{-1}$, in line with the model results at intermediate air temperature (see Fig S6c in the submitted manuscript). And we set $s^* = 0.32$, based on the soil moisture level that marks a substantial reduction in the total leaf-level conductance in water vapor (Fig S2d in the submitted manuscript). These values are also well aligned with literature values (e.g., Laio et al., 2001).

We run the two models in parallel, forced by the same realization of the precipitation input. We determined the cumulated inputs via irrigation; and losses via evapotranspiration, runoff and deep percolation, over periods of 21 days (i.e., the assumed length of anthesis); and the ratios between these quantities and the cumulated precipitation. We tested all the precipitation scenarios included in the original manuscript.

[Figure]

**Figure 1.1: Distribution of 1000 21-day cumulated evapotranspiration (top row), runoff and deep percolation (middle row) and irrigation (bottom row), for rainfed (left) and irrigated cropping (right), for the two soil water balance models: in blue, the outputs of the model used in the original submission, whereby soil water contents above the threshold $s_1$ are lost instantaneously (at the daily time scale); in orange, the outputs of the model assuming a power-law dependence of deep percolation on soil moisture and runoff via saturation excess. Each pair of boxes refers to a different rainfall scenario: from left to right, average total annual precipitation of 500, 700, 900 and 1100 mm, with average event depth of 15 mm (R500, R700, R900 and R1100 respectively); and the two scenarios with a total average annual precipitation of 600 mm, but differing in average precipitation frequency $\lambda_p$ and event depth $\alpha_p$ (Rbaseline, $\alpha_p$=8.2 mm; $\lambda_p$=0.2 d$^{-1}$; Rintermittent, $\alpha_p$=23.5 mm; $\lambda_p$=0.07 d$^{-1}$). The thick horizontal line denotes the median value, the box extends from the first to the third quartile, and the whiskers from the 5$^{th}$ to the 95$^{th}$ percentile. All the other parameters are reported in Table S2 of the original submission. Because a stress avoidance irrigation is implemented, the cumulated evapotranspiration for irrigated cropping (top right) equals the maximum possible evapotranspiration, i.e., 21$ET_{max}$, with no variability induced by the stochasticity of precipitation occurrence.**

[Figure]

**Figure 1.2: Distribution of the ratios of 21-day cumulated evapotranspiration (top row), runoff and deep percolation (middle row) and irrigation (bottom row) to cumulated precipitation, for rainfed (left) and irrigated cropping (right). All the colors and symbols are as in Figure 1.1. The number of ratios is below 1000 when no precipitation was recorded over the 21-day period. The ratio of cumulated evapotranspiration over cumulated precipitation can exceed 1 also in rainfed cropping, because a net reduction in soil water storage can occur over some 21-day periods.**

The dominant soil water balance loss is via evapotranspiration, with deep percolation and runoff playing a secondary role under all precipitation scenarios, and in particular in rainfed agriculture (Figure 1.1). The two soil water balance models lead to similar, albeit not exactly equal, cumulated deep percolation and runoff and its ratio to precipitation (Figure 1.2). An even better match could have been achieved by setting a precipitation-specific threshold $s_1$ such as to limit the already small discrepancies – something we did not attempt in the above analyses; rather we kept the same threshold as in the original submission. The variability across the 21-day periods and precipitation regimes is larger than that stemming from soil water balance model. Because the quantitative importance of deep percolation and runoff is small, the two water balance models yield very similar cumulated evapotranspiration and irrigation, and their ratios to cumulated precipitation (Figure 1.2). The substantial independence of the cumulative ET of model choice is of particular relevance for the results in the manuscript, since evapotranspiration and leaf temperatures are related, all the other conditions being the same.

Other more complex descriptions of the soil water balance, e.g., explicitly considering runoff by infiltration excess and ponding (e.g., Rigby and Porporato, 2006; Manfreda et al., 2010) might indeed lead to wetter soil, by further reducing the amount of water lost to surface runoff. Nevertheless, based on the quantitative analyses above, we expect the resulting difference in soil moisture, and hence on canopy temperature, to be rather small. Further, a slight underestimation of soil water content would lead to conservative estimates of canopy temperature and the risk of canopy temperature exceeding the threshold for potential damage. An even more realistic description of the soil water balance would require abandoning the concept of the bucket model, for a layered soil water balance. Yet, we believe such a detailed description of the soil component would not match the relative simplicity of the description of the soil-plant-atmosphere continuum currently implemented.

In conclusion, these further analyses and considerations support a simplified approach to the modelling of the soil water balance. We would thus prefer not to alter the current soil water balance description, thus maintaining the consistency of temporal scales across the different submodels. In the revisions, we will however further justify our choice, by briefly discussing the results of the above comparison in Section S5.1 Modeling assumptions and their implications. Moreover, as suggested by the Referee, we will add quantitative information on the ratios of cumulated inputs and losses to cumulated precipitation, by means of a figure along the lines of Figure 1.2, but considering only the model used in the manuscript, to be placed in a new section at the beginning of S3 Additional results, devoted to the water fluxes.

Conversely, we believe there is no need to prolong the time of operation of the hydrological processes. Both in the original manuscript and the figures above, we run the model for a series of concatenated 21-day periods, where the conditions at the end of one period are used at the beginning of the subsequent one. In such a way, the conditions at the beginning of each period are fully stochastic, and reflect a long period of operation of all the hydrological processes. In the revised manuscript, we will further emphasise this point in Section S1.5 Numerical simulations.

*The irrigation is triggered whenever soil water potential reached the intervention point which is just above the water stress point of wheat. It would be nice to know what was the resulting average irrigation frequency with this strategy? Is this frequency similar to the typical wheat irrigation frequency? Was the target soil water potential for irrigation optimized to result in typical wheat irrigation frequency? Report also the amount of irrigation and its ratio to precipitation in the result section.*

The target soil moisture was not optimized to result in typical wheat irrigation frequencies, but rather set considering an intervention point corresponding to the soil water potential of incipient water stress for wheat; and a specific irrigation technology. The latter at least dictates the amount of water provided by each irrigation application, with more sophisticated approaches able to provide also extremely small water depths, and cheaper, more commonly employed technologies delivering larger water depths at each application (see, e.g., Vico and Porporato, 2011 and references therein). More in general, the resulting irrigation frequency stems from the combination of intervention point, target level, and all aspects of the water balance, from the precipitation amount and timing to plant water uptake. It is thus difficult to choose parameters to match specific irrigation frequencies, as this would require also altering the intervention point and target level depending on precipitation input. Further, in practical application, the irrigation frequency can also be affected by other aspects, for example access to water according to a specific calendar.

We have however determined the irrigation frequency under the current model parameterization; their distributions under the different precipitation scenarios are summarized in Figure 1.3. There is a minimum period between two subsequent irrigation applications (corresponding to the time necessary for the soil moisture to decline from the target level to the intervention point in the absence of precipitation), and hence a maximum frequency of irrigation. Furthermore, because the temporal resolution of the model is the day, the minimum difference in irrigation frequency is $1/21=0.0476$ d$^{-1}$. As a result, over a period as short as 21 days, the dependence of irrigation frequency on the precipitation regime is small, although the wettest scenarios have lower median irrigation application frequencies.

[Figure]

**Figure 1.3: Distribution of the average irrigation frequencies during 1000 21-day periods. All colors and symbols are as in Figure 1.1.**

Given the number of aspects influencing the irrigation frequency, and to focus on the effects of the statistics of the climatic forcing on canopy temperature, we plan to maintain the choice of the irrigation parameters independent of a specific
location. But, motivated by the Referee's questions, in the revised manuscript, we will 1) clarify the rationale behind the choice of the intervention point for irrigation, linking that to the irrigation technology, in Section 2.2 and S1.3.1; 2) state the average irrigation frequency, under the different precipitation scenarios, in the SI (in the new section at the beginning of S3 Additional results, devoted to the water fluxes). We will also try to place the resulting irrigation frequency in a broader context, with reference to common practice, although, as discussed above, many aspects affect the actual irrigation
frequency.

*Specific comments:*
*L45: "are directly affected high air temperature"*

Thanks for pointing out this typo. The verb will be changed to plural.

*L98: Should you rephrase this question? The results and discussion seem to only report that irrigation cannot completely remove the heat stress but there seems to be no quantification of this effect?*

While we put more emphasis on the inability of irrigation to completely remove the heat stress, Table 1 offers a quantitative measure of the effects of irrigation, by presenting the reduction in the mean fraction of days during anthesis with canopy temperature exceeding the threshold for potential damage from rainfed to irrigated cropping. Nevertheless, the Referee's comment suggests that this aspect is currently not emerging as clearly as it should. In the revised manuscript, we will thus keep the question as is in the introduction, but better clarify the quantification of the role of irrigation, explicitly explaining the meaning of the percentages reported in the Table in the result section; and referring to the third question in the discussion (first paragraph of Section 4.3 Irrigation reduces but does not cancel the risk of heat stress).

*L127: In what sense superficial? Perhaps, surface runoff?*

'Surface' runoff is the correct term. L127 will be modified accordingly.

*L166: What is the basis of the precipitation parameters? Is intermittent case based on some estimate for some region?*

The precipitation parameters were chosen to cover a wide range of regimes, and hence locations, but without reference to a
specific region. We selected the different precipitation timings (and hence event depths) for the same precipitation total in a similar way, and considering that for many regions climate change scenarios predicts a decrease in frequency but an increase in intensity of precipitation, and that such a change could be potentially more damaging to crops than the opposite one.
This approach of selecting precipitation parameters has the advantage of allowing the exploration of a wide range of conditions. Linking the analyses to a specific location (and choosing the corresponding changes in rainfall regime based on
climate change scenarios) would also limit the breath of air temperatures that can be explored. We would thus prefer to maintain the climatic parameter choice independent from specific locations. Yet, we will clarify this aspect in the revised manuscript, specifically in Section 2.2 Case study.

*L298: Reference missing.*

A reference to Table 1 should have appeared here.

**References**

Clapp, R. B., and Hornberger, G. M.: Empirical equations for some soil hydraulic properties, Water Resour Res, 14, 601-604, https://doi.org/10.1093/jxb/eri174, 1978.
Laio, F., Porporato, A., Ridolfi, L., and Rodriguez-Iturbe, I.: Plants in water-controlled ecosystems: active role in hydrologic
processes and response to water stress - II. Probabilistic soil moisture dynamics, Adv Water Resour, 24, 707-723, https://doi.org/10.1016/s0309-1708(01)00005-7, 2001.
Manfreda, S., Scanlon, T. M., and Caylor, K. K.: On the importance of accurate depiction of infiltration processes on modelled soil moisture and vegetation water stress, Ecohydrology, 3, 155-165, https://doi.org/10.1002/eco.79, 2010.
Rigby, J. R., and Porporato, A.: Simplified stochastic soil-moisture models: a look at infiltration, Hydrol. Earth Syst. Sci.,
10, 861-871, 10.5194/hess-10-861-2006, 2006.
Vico, G., and Porporato, A.: From rainfed agriculture to stress-avoidance irrigation: I. A generalized irrigation scheme with stochastic soil moisture, Adv Water Resour, 34, 263-271, https://doi.org/10.1016/j.advwatres.2010.11.010, 2011.

---

## Referee Comment (RC2) · Marijn van der Velde (Referee) · 22 Dec 2020

Review of Canopy temperature and heat stress are increased by compound high air temperature and water stress, and reduced by irrigation – A modeling analysis by Xiangyu Luan, Giulia Vico as submitted to HESSD

The manuscript submitted by Luan and Vico investigates a relevant topic and does so using a newly developed mechanistic model that allows to develop a better understanding in the processes and feedbacks that determine the coupled impact of water

and heat stress in irrigated crops. Irrigation can alleviate water stress but can also lower the maximum canopy temperature and period of heat stress experienced during heat waves.

The manuscript focuses on presenting the model and benchmarks its performance in a case study for wheat. Along with the manuscript comes an excellent and extensive model description included in the supplemental material. I advise the authors to eventually publish their full model and code in a citable open source repository.

I commend the authors for focusing on transparency and simplicity, for instance in defining the crop and phenology stage specific threshold temperatures that would trigger damage from heat stress. The authors explicitly consider the stochastic effects of temperature and precipitation. While I appreciate the illustration of the model in the case for a hypothetical wheat crop, I do look forward to further scrutiny of the model against data from field experiments. A next challenge will be to untangle the net effects on biomass and yield. By lowering temperature, irrigation is also delaying harvesting and thus allowing for a longer grain filling period. The research also has implications for water management in spelling out the relative benefits and limitations of irrigation used specifically for cooling during heat waves. This was lacking previously (e.g. Van der Velde et al., 2009).

One point of clarification needs to be made with regard to soil water balance and effective rooting depth. Research has shown that deeper rooting vegetation and thus access to soil moisture can lead to contrasting responses of vegetation and canopy temperature to heatwaves (e.g. see work of Teuling et al., 2010, but also Zaitchik et al., 2007). Have you done sensitively test of your model with respect to effective rooting depth parameter (now 0.3 m)? While not necessary to consider for this manuscript, you may detail this a bit further by referring to your previous work.

L298 Reference missing

References (included for information)

Saadia, R., Huber, L., Lacroix, B., 1996. Modification du microclimat d'un couver de mais au moyen de l'irrigation par aspersion en vue de la gestion des stress thermiques des organes reproducteurs. Agronomie 16, 465–477.

Teuling A.J., Seneviratne S.I., Stockli R., Reichstein M., Moors E., Ciais P., Luyssaert S., (...), Wohlfahrt G., 2010. Contrasting response of European forest and grassland energy exchange to heatwaves Nature Geoscience, 3 (10), 722-727.

Van der Velde, M., Wriedt, G., Bouraoui, G., 2010. Estimating irrigation use and effects on maize yield during the 2003 heatwave in France. Agriculture, Ecosystems & Environment, 135, 90-97, doi:10.1016/j.agee.2009.08.017.

Zaitchik, B.F., Macalady, A.K., Bonneau, L.R., Smith, R.B., 2006. Europe's 2003 heat wave: a satellite view of impacts and land-atmosphere feedbacks. Int. J. Climatol. 26, 743–769.

---

## Author Comment (AC2) · 14 Jan 2021

https://doi.org/10.5194/hess-2020-549

**Response to Marijn van der Velde (Referee # 2)**

**In italics the original comments; in normal font our responses.**

*The manuscript submitted by Luan and Vico investigates a relevant topic and does so using a newly developed mechanistic model that allows to develop a better understanding in the processes and feedbacks that determine the coupled impact of water and heat stress in irrigated crops. Irrigation can alleviate water stress but can also lower the maximum canopy temperature and period of heat stress experienced during heat waves.*

*The manuscript focuses on presenting the model and benchmarks its performance in a case study for wheat. Along with the manuscript comes an excellent and extensive model description included in the supplemental material. I advise the authors to eventually publish their full model and code in a citable open source repository.*

*I commend the authors for focusing on transparency and simplicity, for instance in defining the crop and phenology stage specific threshold temperatures that would trigger damage from heat stress. The authors explicitly consider the stochastic*
*effects of temperature and precipitation.*

We thank Dr. van der Velde for the very supportive comments. For enhanced transparency and to facilitate further applications of the model, we will consider uploading our codes to an open-source repository (e.g., Zenodo).

*While I appreciate the illustration of the model in the case for a hypothetical wheat crop, I do look forward to further scrutiny of the model against data from field experiments.*

We consider a thorough comparison with observations beyond the scope of the current contribution. Such a comparison would require selecting a specific location, for which data for model calibration, environmental conditions, and canopy
temperatures over several seasons or irrigation treatments are available. While some datasets are available (e.g., 'T-FACE-Maricopa'/Hot Serial Cereals and 'China wheat'; (Webber et al., 2018)), most results reported in the literature focus on the relations between leaf water potential to canopy-to-air temperature difference (which is affected by environmental conditions, including vapor pressure deficit). Even in the presence of irrigation treatments, information on the soil water availability is often not reported. For these reasons, we had qualitatively checked our model's results by comparing its
estimates with field observations and other model results in Section 4.1 (Soil water availability and air temperature jointly affect canopy temperature). There, we had focused on canopy-to-air temperature differences under well-watered or water-limited conditions, to reduce the effects of the specific conditions. Should we be offered the possibility to revise the manuscript, we will further extend this comparison, based on a deeper literature review.

*A next challenge will be to untangle the net effects on biomass and yield.*

The net effects of compound heat and water stress on biomass and yields are indeed extremely interesting… and difficult to include in a process-based model, due to their complexity and high level of uncertainty in many processes and their parameterization. These difficulties explain why the compound effects of temperature and precipitation (and hence water
availability) have received less attention in general than single abiotic stresses (Rötter et al., 2018). When considered, the role of co-occurring stresses have most often been determined based on statistical models (e.g., Carter et al., 2016; Matiu et al., 2017) but more rarely included in process-based models (Rezaei et al., 2015). Further, even heat stress in isolation has been seldom accounted for in models (Barlow et al., 2015). As also implicitly suggested by the Referee, these aspects deserve stand-alone future contributions. Nevertheless, we plan to add a short paragraph in Section 4.3 (Irrigation reduces
but does not cancel the risk of heat stress) in a revised submission, to clarify the linkages between canopy temperatures and final yields.

*By lowering temperature, irrigation is also delaying harvesting and thus allowing for a longer grain filling period.*

This is an excellent point. For simplicity, we had chosen and explicitly mentioned (L148) that the length of the anthesis period was independent of the air temperature scenario. But also irrigation could alter the length of any stage of the phenology, thanks to its cooling effects, all else being the same. We plan to mention this aspect in Section 2.2 Case study when revising the manuscript.

*The research also has implications for water management in spelling out the relative benefits and limitations of irrigation used specifically for cooling during heat waves. This was lacking previously (e.g. Van der Velde et al., 2009).*

In Section 4.3 (Irrigation reduces but does not cancel the risk of heat stress), we had briefly explained the benefits and limitations of irrigation as a way to reduce the occurrence of heat stress in crops, but also its costs and implications in terms
of water requirements and negative environmental consequences. In a revised manuscript, we plan to deepen this discussion by completing Section 4.3 explicitly mentioning the effects of extensive irrigation at the regional scale; and the corresponding effects of a change in temperature regime mentioned by the Referee in the previous comment. Altogether, these direct and indirect effects can significantly alter the final crop yield, as for example clearly shown in the article suggested by the Referee.

*One point of clarification needs to be made with regard to soil water balance and effective rooting depth. Research has shown that deeper rooting vegetation and thus access to soil moisture can lead to contrasting responses of vegetation and canopy temperature to heatwaves (e.g. see work of Teuling et al., 2010, but also Zaitchik et al., 2007). Have you done sensitively test of your model with respect to effective rooting depth parameter (now 0.3 m)? While not necessary to consider*
*for this manuscript, you may detail this a bit further by referring to your previous work.*

As long as the effective rooting zone remains far from the water table, its depth, $Z_r$, has multiple effects on plant water availability (and hence canopy temperature; see Fig. 2 in the original submission). These effects depend on precipitation regime (see e.g. Fig. 8 and 10 in Laio et al., 2001, for the combined effects of rooting depth and climatic conditions on the
probability distribution of soil moisture and the averaged soil water balance), as well as plant water use strategy. Assuming the same plant water use strategy (i.e., response to decreasing soil water potential), deeper roots are expected to reduce losses by runoff and deep percolation and stabilize soil moisture.
Nevertheless, by considering a range of $Z_r$ compatible with rooting depths for wheat (and annual crops in general; Jackson et al., 1996), we found rooting depth had no appreciable effect on mean canopy temperatures, for the baseline and more
intermittent rainfall scenarios (Figure 2.1). The small difference in $Z_r$ and the fact that this was the only parameter changed in this sensitivity analysis likely explain why these results are in contrast with those reported in the papers suggested by the Referee. There, different land uses are associated not only to likely differences in rooting depth, with forests having deeper roots than grasslands and croplands, but also with different water use strategies, with forests generally having a more conservative water use, i.e., a stronger regulation of stomatal conductance in response to temperature, solar radiation, and
vapor pressure deficit, or different coupling to the atmosphere (see, e.g., the interpretation proposed by Teuling et al., 2010). Indeed, in three wheat varieties, cooler canopies at anthesis were correlated with deeper roots, but also higher canopies (Li et al., 2019), making it difficult to disentangle the effect of rooting depth alone based on experimental observations. The number and timing of processes and feedbacks involved in defining the differential response of forests and grasslands in the face of climatic conditions could also explain the contrasting results emerging from the two papers suggested by the Referee.
Given the negligible effect of $Z_r$ in isolation when focusing on realistic ranges for wheat and considering that the manuscript is currently focusing on the effects of changes in the pedoclimatic conditions (i.e., assuming a set crop and its features), we would prefer not to add this dimension to a revised manuscript. Yet, we will briefly touch upon this in Section 3. Results, with reference to previous results.

[Figure]

**Figure 2.1: Mean canopy temperature during anthesis ($T_{c,mean}$), for two effective rooting depths $Z_r$ (x-axis) and different precipitation (colors). R$_{baseline}$ and R$_{intermittent}$ represent rainfed cropping, respectively under baseline precipitation ($\alpha_p$=8.2 mm; $\lambda_p$=0.2 d$^{-1}$) and more intermittent precipitation ($\alpha_p$=23.5 mm; $\lambda_p$=0.07 d$^{-1}$). For each climatic and rooting depth scenario, 100 21-day simulations were run. The horizontal black lines are the median**
**values; the boxes extend from the first to the third quartile; whiskers cover the whole range.**

*L298 Reference missing*

A reference to Table 1 should have appeared here but the link got corrupted. It will be added in the revised manuscript.

*References (included for information)*
*Saadia, R., Huber, L., Lacroix, B., 1996. Modification du microclimat d'un couver de mais au moyen de l'irrigation par*
*aspersion en vue de la gestion des stress thermiques des organes reproducteurs. Agronomie 16, 465–477.*
*Teuling A.J., Seneviratne S.I., Stockli R., Reichstein M., Moors E., Ciais P., Luyssaert S., (...), Wohlfahrt G., 2010.*
*Contrasting response of European forest and grassland energy exchange to heatwaves Nature Geoscience, 3 (10), 722-727.*
*Van der Velde, M., Wriedt, G., Bouraoui, G., 2010. Estimating irrigation use and effects on maize yield during the 2003*
*heatwave in France. Agriculture, Ecosystems & Environment, 135, 90-97, doi:10.1016/j.agee.2009.08.017.*
*Zaitchik, B.F., Macalady, A.K., Bonneau, L.R., Smith, R.B., 2006. Europe's 2003 heat wave: a satellite view of impacts and*
*land-atmosphere feedbacks. Int. J. Climatol. 26, 743–769.*

*L298: Reference missing.*

**120 References**

Barlow, K. M., Christy, B. P., O'Leary, G. J., Riffkin, P. A., and Nuttall, J. G.: Simulating the impact of extreme heat and frost events on wheat crop production: A review, Field Crop Res, 171, 109-119, https://doi.org/10.1016/j.fcr.2014.11.010, 2015.

Carter, E., Melkonian, J., Riha, S., and Shaw, S.: Separating heat stress from moisture stress: analyzing yield response to
high temperature in irrigated maize, Env Res Lett, 11, 094012, https://doi.org/10.1088/1748-9326/11/9/094012, 2016.

Jackson, R. B., Canadell, J., Ehleringer, J. R., Mooney, H. A., Sala, O. E., and Schulze, E. D.: A global analysis of root distributions for terrestrial biomes, Oecologia, 108, 389-411, https://doi.org/10.1007/BF00333714, 1996.

Laio, F., Porporato, A., Ridolfi, L., and Rodriguez-Iturbe, I.: Plants in water-controlled ecosystems: active role in hydrologic processes and response to water stress - II. Probabilistic soil moisture dynamics, Adv Water Resour, 24, 707-723,
https://doi.org/10.1016/s0309-1708(01)00005-7, 2001.

Li, X., Ingvordsen, C. H., Weiss, M., Rebetzke, G. J., Condon, A. G., James, R. A., and Richards, R. A.: Deeper roots associated with cooler canopies, higher normalized difference vegetation index, and greater yield in three wheat populations grown on stored soil water, J Exp Bot, 70, 4963-4974, https://doi.org/10.1093/jxb/erz232, 2019.

Matiu, M., Ankerst, D. P., and Menzel, A.: Interactions between temperature and drought in global and regional crop yield
variability during 1961-2014, Plos One, 12, https://doi.org/10.1371/journal.pone.0178339, 2017.

Rezaei, E. E., Webber, H., Gaiser, T., Naab, J., and Ewert, F.: Heat stress in cereals: mechanisms and modelling, Eur J Agron, 64, 98-113, https://doi.org/10.1016/j.eja.2014.10.003, 2015.

Rötter, R. P., Appiah, M., Fichtler, E., Kersebaum, K. C., Trnka, M., and Hoffmann, M. P.: Linking modelling and experimentation to better capture crop impacts of agroclimatic extremes-A review, Field Crop Res, 221, 142-156,
https://doi.org/10.1016/j.fcr.2018.02.023, 2018.

Teuling, A. J., Seneviratne, S. I., Stöckli, R., Reichstein, M., Moors, E., Ciais, P., Luyssaert, S., van den Hurk, B., Ammann, C., Bernhofer, C., Dellwik, E., Gianelle, D., Gielen, B., Grünwald, T., Klumpp, K., Montagnani, L., Moureaux, C., Sottocornola, M., and Wohlfahrt, G.: Contrasting response of European forest and grassland energy exchange to heatwaves, Nature Geoscience, 3, 722-727, 10.1038/ngeo950, 2010.

Webber, H., White, J. W., Kimball, B. A., Ewert, F., Asseng, S., Rezaei, E. E., Pinter, P. J., Hatfield, J. L., Reynolds, M. P., Ababaei, B., Bindi, M., Doltra, J., Ferrise, R., Kage, H., Kassie, B. T., Kersebaum, K. C., Luig, A., Olesen, J. E., Semenov, M. A., Stratonovitch, P., Ratjen, A. M., LaMorte, R. L., Leavitt, S. W., Hunsaker, D. J., Wall, G. W., and Martre, P.: Physical robustness of canopy temperature models for crop heat stress simulation across environments and production conditions, Field Crop Res, 216, 75-88, https://doi.org/10.1016/j.fcr.2017.11.005, 2018.

---

## Author Response (AR2)

**Luan, X. and Vico, G.: Canopy temperature and heat stress are increased by compound high air temperature and water stress, and reduced by irrigation – A modeling analysis, Hydrol. Earth Syst. Sci. Discuss. [preprint], https://doi.org/10.5194/hess-2020-549, in review, 2020.**

**List of changes – January 2021**

Below we list the changes implemented in response to the comments and suggestions of the Referees. The rationale of such changes is explained in detail in our previously submitted responses (available at https://hess.copernicus.org/preprints/hess-2020-549/).

Beyond the specific changes listed below, the manuscript and Supplementary Information (SI) have been carefully proof-read. Some minor changes not specifically requested by the Referees have been implemented in the text for enhanced clarity or readability.

All the implemented changes are apparent in the version including the tracked changes uploaded on the manuscript portal. The points below follow the same order of those taken up in our responses to the Referees. Line and section numbers refer to the revised version of the manuscript.

**In response to the comments and suggestions of Referee #1, and in line with our previously submitted response to them, we have implemented the following changes in the revised manuscript:**

1) Added a paragraph justifying the modeling approach to the soil water balance and discussing its key simplifying assumptions in Section S5.1 Modeling assumptions and their implications (SI, L421-433).

2) Added a new section at the beginning of Section S3 Additional results, titled S3.1 Water fluxes, presenting three new figures (Fig. S3-S5) summarizing the ratios of cumulated losses and irrigation to cumulated precipitation, for the model used in the manuscript. Because the addition of these pieces of information on the water fluxes required re-running the model, we have also replaced the corresponding figures (Fig. 4 and 5 in the main text; Fig. S6-S7 in the SI) and re-made the statistical analyses (SI, Table S3-S8), so that all the results are based on the exact simulations. The number of simulations was already high so using a new set of simulations did not lead to appreciable change in the results.

3) In Section S1.5 Numerical simulations, further emphasized the fact that we run the model for a series of concatenated 21-day periods, where the conditions at the end of one period are used at the beginning of the subsequent one, thus reflecting a long period of operation of all the hydrological processes (SI, L243-249).

4) Clarified the rationale behind the choice of the intervention point for irrigation, linking that to the irrigation technology, in Section 2.3 (L179-181) and S1.3.1 (SI, L187-194).

5) Reported and discussed the irrigation frequency, under the different climatic scenarios, in the new section at the beginning of S3 Additional results, devoted to the water fluxes (Section S3.1, L300-305). As discussed in the response to the Referee, the focus on such a short period as the anthesis, and the joint effects of pedoclimatic

conditions and crop features, make it difficult to compare the resulting modelled frequency of irrigation and typical applications. We however note that the irrigation application depths are in line with those typical of traditional acclimation.

6) Clarified our quantification of the effectiveness of irrigation in reducing canopy temperature and the duration of potentially damaging conditions, by further specifying the corresponding question in the introduction (L83-84); and referring to Table 1 and explaining its content in the discussion (Section 4.3 Irrigation reduces but does not cancel the risk of heat stress; L304-308).

7) Clarified that, while the model could be forced with site-specific climatic conditions, we used a weather generator and varied its parameters, in order to effectively and systematically explore a wide range of conditions in Section 2.3 Case study (L158-160).

8) Corrected the typos the Referee kindly pointed us to; and added the missing reference to Table 1 in Section 4.3.

**In response to the comments and suggestions of Referee # 2, and in line with our previously submitted response to them we have implemented the following changes in the revised manuscript:**

9) Started to prepare the model code for submission to Zenodo (https://zenodo.org/) upon manuscript acceptance.

10) Extended our review of the literature reporting differences in canopy-to-air temperatures under well-watered and water-stressed conditions in Section 4.1 (Soil water availability and air temperature jointly affect canopy temperature), to provide further evidence of the realism of the model results (L255-259).

11) Added a short paragraph in Section 4.3 (Irrigation reduces but does not cancel the risk of heat stress) to clarify the linkages between canopy temperatures and final yields, exploiting existing literature data (L301-304).

12) Mentioned the potential effects of irrigation on the duration of the developmental stages (including anthesis) in Section 2.3 Case study (L151-152) and Section 4.3 (L312-313).

13) Deepened the discussion on the effects of irrigation in Section 4.3 by explicitly mentioning the effects of extensive irrigation at the regional scale; and the corresponding effects of a change in temperature regime (L311-314).

14) Mentioned the role of rooting depth in Section 3. Results, with reference to previous result (L236-239).

15) Corrected the reference to Table 1 in Section 4.3.